# Human Sinoatrial Node Pacemaker Activity: Role of the Slow Component of the Delayed Rectifier K^+^ Current, I_Ks_

**DOI:** 10.3390/ijms24087264

**Published:** 2023-04-14

**Authors:** Arie O. Verkerk, Ronald Wilders

**Affiliations:** 1Department of Medical Biology, Amsterdam Cardiovascular Sciences, Amsterdam UMC, University of Amsterdam, 1105 AZ Amsterdam, The Netherlands; a.o.verkerk@amsterdamumc.nl; 2Department of Experimental Cardiology, Heart Center, Amsterdam Cardiovascular Sciences, Amsterdam UMC, University of Amsterdam, 1105 AZ Amsterdam, The Netherlands

**Keywords:** slow delayed rectifier potassium current, human, sinoatrial node, patch clamp, action potential clamp, computer simulations, heart rate, β-adrenergic stimulation

## Abstract

The pacemaker activity of the sinoatrial node (SAN) has been studied extensively in animal species but is virtually unexplored in humans. Here we assess the role of the slowly activating component of the delayed rectifier K^+^ current (I_Ks_) in human SAN pacemaker activity and its dependence on heart rate and β-adrenergic stimulation. HEK-293 cells were transiently transfected with wild-type *KCNQ1* and *KCNE1* cDNA, encoding the α- and β-subunits of the I_Ks_ channel, respectively. KCNQ1/KCNE1 currents were recorded both during a traditional voltage clamp and during an action potential (AP) clamp with human SAN-like APs. Forskolin (10 µmol/L) was used to increase the intracellular cAMP level, thus mimicking β-adrenergic stimulation. The experimentally observed effects were evaluated in the Fabbri–Severi computer model of an isolated human SAN cell. Transfected HEK-293 cells displayed large I_Ks_-like outward currents in response to depolarizing voltage clamp steps. Forskolin significantly increased the current density and significantly shifted the half-maximal activation voltage towards more negative potentials. Furthermore, forskolin significantly accelerated activation without affecting the rate of deactivation. During an AP clamp, the KCNQ1/KCNE1 current was substantial during the AP phase, but relatively small during diastolic depolarization. In the presence of forskolin, the KCNQ1/KCNE1 current during both the AP phase and diastolic depolarization increased, resulting in a clearly active KCNQ1/KCNE1 current during diastolic depolarization, particularly at shorter cycle lengths. Computer simulations demonstrated that I_Ks_ reduces the intrinsic beating rate through its slowing effect on diastolic depolarization at all levels of autonomic tone and that gain-of-function mutations in *KCNQ1* may exert a marked bradycardic effect during vagal tone. In conclusion, I_Ks_ is active during human SAN pacemaker activity and has a strong dependence on heart rate and cAMP level, with a prominent role at all levels of autonomic tone.

## 1. Introduction

The pacemaker activity of the sinoatrial node (SAN) has been studied extensively in animal species but is virtually unexplored among humans [1]. According to the different heart rates, with high rates in small mammals and slower rates in humans [2,3,4], differences in the underlying pacemaker activity mechanism between animal species and humans can be expected [5,6]. Diseases and remodeling of the SAN [7,8,9,10,11], inherited cardiac syndromes [12,13,14,15,16], and drugs [17,18,19,20,21] frequently lead to changes in SAN pacemaker formation and/or sick sinus syndrome. A better understanding of the basic mechanisms of human SAN automaticity is therefore clinically important [8,17,22].

Chandler et al. [23] characterized the ‘molecular architecture’ of the human SAN based on the messenger RNA (mRNA) levels of 120 ion channels and the expression level of some related proteins, and they concluded that the observed expression pattern of these ion channels is “appropriate to explain pacemaking”. We actually measured action potentials (APs) of single pacemaker cells isolated from a human SAN using a patch clamp methodology [24]. We recorded spontaneous APs with a clear, but slow, diastolic depolarization phase resulting in an intrinsic cycle length of ≈830 ms (72 beats per min (bpm)). In addition, we found in these single human SAN cells a hyperpolarization-activated current (I_f_) [24], a fast and large inward current with characteristics of the fast Na^+^ current (I_Na_) [25], and a spontaneous Ca^2+^ transient [26]. The role of Ca^2+^ transients in human SAN automaticity was further studied in detail by Tsutsui et al. [27] in freshly isolated SAN cells from the donor hearts of four patients without a history of major cardiovascular diseases. They demonstrated that spontaneous and rhythmic local Ca^2+^ releases, generated by the so-called ‘Ca^2+^ clock’ [28,29], are coupled to electrogenic surface membrane molecules (the ‘membrane clock’ or ‘M clock’ [28,29]) to trigger spontaneous APs, and that cAMP protein kinase A (PKA) signaling regulates the clock coupling [27]. However, many exact details about single human SAN cell electrophysiology are still unknown due to the extreme difficulty of obtaining freshly isolated human SAN cells for in vitro studies [5,30].

In their characterization of the ‘molecular architecture’ of the human SAN, Chandler et al. [23] found mRNA for the KvLQT1 channel, which is responsible for the delayed rectifier K^+^ current (I_Ks_) and is also known as KCNQ1 or Kv7.1 channel. I_Ks_ channels are formed by pore-forming KCNQ1 (Kv7.1) α-subunits and KCNE1 (minK) β-subunits, which are encoded by the *KCNQ1* and *KCNE1* genes, respectively [31,32]. I_Ks_ is a voltage-gated, outwardly directed repolarizing membrane current that activates at potentials positive to −50 mV. It deactivates upon repolarizing steps [33], and it may accumulate at high stimulation rates because of its slow deactivation kinetics [34,35]. I_Ks_ is modulated by various factors (for reviews, see Wang et al. [36], Wu and Larsson [37], and Sanguinetti and Seebohm [38]), including β-adrenergic stimulation [39], making this membrane current increase during exercise. The role of I_Ks_ in working myocardium APs and ventricular channelopathies is well established [33,37,40], but its function in pacemaker cells is less clear. On the one hand, one may argue that I_Ks_ will be active during the AP phase and that an increase in I_Ks_ increases the pacing rate of SAN cells through a shortening effect of this outward current on AP duration. On the other hand, one may hypothesize that I_Ks_ will be active during diastolic depolarization and that an increase in I_Ks_ decreases the pacing rate by exercising a suppressing effect on diastolic depolarization.

In the present study, we investigated the mechanism by which I_Ks_ modulates human SAN pacemaker activity. We first characterized the biophysical properties of KCNQ1/KCNE1 channels transfected in HEK-293 cells in absence and presence of forskolin, which was added to increase the intracellular cAMP level and thus mimic β-adrenergic stimulation conditions. In these conventional, square-step voltage clamp experiments, β-adrenergic stimulation activated KCNQ1/KCNE1 current by an increase in its conductance as well as a negative shift in its steady-state activation curve, thereby increasing the degree of current activation at any given voltage. Secondly, we measured KCNQ1/KCNE1 current during human SAN-like AP waveforms in absence and presence of forskolin, and at various cycle lengths. In these AP clamp experiments, we found that KCNQ1/KCNE1 current during AP repolarization was substantial and only slightly dependent on cycle length, but significantly increased in the presence of forskolin. The KCNQ1/KCNE1 current during diastolic depolarization, on the other hand, was relatively small, but increased at shorter cycle lengths as well as in the presence of forskolin. To study the functional role of the *KCNQ1*/*KCNE1* encoded I_Ks_ in human SAN pacemaker activity, we modified the I_Ks_ equations of the Fabbri–Severi computer model of an isolated human SAN pacemaker cell [41] according to our patch-clamp data and ran the thereby obtained model at different levels of autonomic tone, also studying the effects of a gain-of-function mutation in *KCNQ1*. We found that I_Ks_ reduces the intrinsic beating rate through its slowing effect on diastolic depolarization at all levels of autonomic tone and that gain-of-function mutations in *KCNQ1* may exert a marked bradycardic effect during vagal tone. A preliminary version of this study was presented at the 49th Computing in Cardiology conference in Tampere, Finland, and published in the conference proceedings [42].

## 2. Results

### 2.1. KCNQ1/KCNE1 Current in Response to Square Voltage Clamp Steps

Figure 1, A–F summarizes the basic biophysical characteristics of ion currents flowing through KCNQ1/KCNE1 channels expressed in HEK-293 cells. Figure 1A shows typical KCNQ1/KCNE1 currents in a HEK-293 cell under control conditions (left traces) and in the presence of 10 µmol/L forskolin (right traces). The HEK-293 cell was voltage-clamped from a holding potential of −80 mV to various test potentials (ranging from −80 to +110 mV, in 10 mV increments) for 2 s, as diagrammed in the protocol at the top of Figure 1A. The depolarizing steps were followed by a 2-s step to −60 mV to record tail currents. As typical for a KCNQ1/KCNE1 current, the threshold of activation was approximately −50 mV and the current amplitude progressively increased with more depolarized potentials. Figure 1B shows the average current-voltage (I-V) relationships of seven cells. Forskolin significantly increased the KCNQ1/KCNE1 current density over the entire voltage range from −10 to +110 mV. At +110 mV, forskolin increased the KCNQ1/KCNE1 current density from 644 ± 60 to 839 ± 72 pA/pF.

The KCNQ1/KCNE1 current activated with a characteristic time course. Typically, KCNQ1/KCNE1 current activated faster at more positive potentials (Figure 1A). The time course of activation was well-fitted by a mono-exponential function, characterized by a single time constant (τ) of activation. The filled squares of Figure 1E show the thus obtained time constant of activation plotted against the depolarizing test potential in absence and presence of forskolin. Forskolin significantly increased the KCNQ1/KCNE1 current activation rate by ≈40% over the entire voltage range from −10 to +110 mV.

The normalized amplitude of the tail current at −60 mV is plotted against the preceding depolarizing test potential in Figure 1C to determine the voltage dependence of the amount of KCNQ1/KCNE1 current activation in the absence and the presence of forskolin. Forskolin shifted the thus obtained steady-state activation curve by −15.3 ± 3.0 mV (*n* = 7; *p* < 0.01), thus increasing the voltage range over which the KCNQ1/KCNE1 current is active during a SAN AP. As shown in Figure 1C (inset), the half-maximal activation voltage (V_½_) changed from 7.8 ± 3.7 mV under control conditions to −7.6 ± 2.7 mV in the presence of forskolin (*n* = 7; *p* < 0.01). The slope factor (k) of the steady-state activation curve was not significantly affected by forskolin (27.5 ± 2.7 mV (control) versus 24.1 ± 2.7 mV (forskolin)).

KCNQ1/KCNE1 currents that were activated during depolarization showed deactivation upon subsequent repolarization. Figure 1D shows a typical example of deactivation in a HEK-293 cell in the absence and the presence of forskolin. The cell was first voltage-clamped from a holding potential of −80 mV to +100 mV for 1 s to ensure full activation. This depolarizing step was followed by a 2-s step to test potentials ranging from −120 to −20 mV, in 10 mV increments, as diagrammed in the protocol at the top of Figure 1D. As typical for KCNQ1/KCNE1 currents, deactivation became faster at more negative test potentials (Figure 1D). The time course of deactivation was well-fitted by a mono-exponential function, characterized by a single time constant of deactivation. The filled circles of Figure 1E show the thus-obtained time constant of deactivation plotted against the test potential in the absence and the presence of forskolin. Forskolin did not affect the KCNQ1/KCNE1 current deactivation rate over the entire voltage range from −120 to −20 mV.

The fully activated I-V relationship was analyzed by determining the peak of the deactivating tail current. The fully activated I-V relationship was linear in the absence as well as in the presence of forskolin (Figure 1F). In each case, the reversal potential was close to the Nernst potential for K^+^. The reversal potential, determined from linear fits to the individual I-V relationships of the seven cells, was −86.4 ± 6.6 and −86.5 ± 6.4 mV under control conditions and in the presence of forskolin, respectively (Figure 1F, upper inset). The fully activated conductance, determined from the slopes of the linear fits, increased significantly by 23 ± 3% from 4.0 ± 0.3 nS/pF under control conditions to 4.9 ± 0.4 nS/pF in the presence of forskolin (Figure 1F, lower inset).

### 2.2. KCNQ1/KCNE1 Current in Response to Human SAN-like AP Waveforms

The square-step voltage clamp experiments described in Section 2.1 above are very useful to determine the basic biophysical properties of the KCNQ1/KCNE1 current. However, despite the detailed knowledge of its kinetic properties, the activity of the KCNQ1/KCNE1 current during the various phases of a human SAN cell AP, in particular during diastolic depolarization, remains difficult to predict. Therefore, we carried out AP clamp experiments with command potentials based on the AP waveform of an isolated human SAN cell, as set out in Section 4.4.

First, we assessed the KCNQ1/KCNE1 current characteristics during a train of human SAN-like AP waveforms with a cycle length of 750 ms (80 bpm; Figure 2A, top). Figure 2, A (bottom) and B, shows the average KCNQ1/KCNE1 current characteristics of six HEK-293 cells. The KCNQ1/KCNE1 current increases during the upstroke of the human SAN-like AP, likely due to activation of the channels and an increase in the driving force of the K^+^ current. The KCNQ1/KCNE1 current reaches its peak during the repolarization phase of the AP, near 0 mV, and subsequently decreases in amplitude during the remaining part of the AP repolarization phase, likely due to the diminishing driving force of the K^+^ current and deactivation of the channels. Near the maximum diastolic potential (MDP) of ≈−64 mV and during the early phase of the diastolic depolarization (from the MDP towards −55 mV), some current is still active (Figure 2B, inset). The current reaches its minimum near −55 mV, but does not become zero, indicating that not all channels deactivated and/or that some deactivated channels already (re)activated. In Figure 2C, we plotted the average fully activated current (dashed line), constructed by extrapolating the currents measured at +80 mV and more positive with the square-step voltage clamp protocol of Figure 1A (Figure 2C, black-filled circles), and the currents active during the human SAN-like AP waveform (Figure 2C, solid line) in the same six cells. For comparison, the gray-filled circles show the maximum current at voltages ranging from −80 to +70 mV as determined with the square-step voltage clamp protocol. The KCNQ1/KCNE1 current measured during the human SAN-like AP waveform was maximally ≈25% of the fully activated current (at a membrane potential near 0 mV; Figure 2C, arrows), indicating that maximally ≈25% of the channels will open during a human SAN-like AP waveform with a cycle length of 750 ms.

Next, we assessed the effects of forskolin on KCNQ1/KCNE1 current during human SAN-like AP waveforms with cycle lengths ranging from 500 to 1500 ms (40–120 bpm), thus studying the dependence of KCNQ1/KCNE1 current on cAMP levels and cycle length. Figure 3, A and B shows the effects of 10 µmol/L forskolin on the average KCNQ1/KCNE1 current (*n* = 6) at a cycle length of 750 ms (80 bpm). In the presence of forskolin, KCNQ1/KCNE1 current increases more quickly during the upstroke of the human SAN-like AP than in the absence of forskolin, in line with the decrease in the activation time constant of Figure 1E. Furthermore, the KCNQ1/KCNE1 current is substantially larger in the presence of forskolin (see also below), consistent with the square-step voltage clamp experiments of Figure 1. Interestingly, in the presence of forskolin, KCNQ1/KCNE1 current is clearly active during the entire diastolic depolarization phase (Figure 3A, bottom; Figure 3B, inset), which can be attributed to the ≈−15 mV shift of the steady-state activation curve in response to forskolin (Figure 1C).

We also studied the KCNQ1/KCNE1 current activity at shorter (500 and 600 ms) as well as longer (1000 and 1500 ms) cycle lengths. Figure 3, C and D shows the effects of changes in cycle length in the absence of forskolin, whereas results obtained in the presence of forskolin are shown in Figure 3, E and F. The average KCNQ1/KCNE1 current of Figure 3C and the phase plane plot of Figure 3D both suggest that the peak current reached during AP repolarization in the absence of forskolin slightly decreases with increasing cycle length. Accordingly, the peak current at a cycle length of 1500 ms differed significantly, albeit to a small extent, from that at cycle lengths of 500, 600, and 1000 ms (*n* = 6), as indicated in Figure 4A (green bars). This statistical significance may seem at odds with the error bars of Figure 4A, but it should be realized that all data are ‘paired’, both with respect to absence or presence of forkolin and with respect to cycle length, so that a powerful two-way Repeated Measures (RM) ANOVA test could be used.

Both Figure 3C and Figure 3D (inset) also suggest that the current during diastolic depolarization in the absence of forskolin decreases with increasing cycle length, becoming approximately zero at cycle lengths of 1000 and 1500 ms. Accordingly, the current at −60 mV during diastolic depolarization at a cycle length of 500 ms differed significantly from that at cycle lengths of 1000 and 1500 ms (*n* = 6), as indicated in Figure 4B (green bars). The latter data demonstrate that at faster heart rates, the KCNQ1/KCNE1 current may ‘accumulate’ and thus increase during the diastolic depolarization phase, likely because of incomplete deactivation.

Figure 3, E and F shows the cycle length dependence of the KCNQ1/KCNE1 current in the presence of forskolin. The average KCNQ1/KCNE1 current of Figure 3E and the phase- plane plot of Figure 3F both suggest that the peak current reached during AP repolarization in the presence of forskolin slightly decreases with increasing cycle length, as in the absence of forskolin. Statistical significance is limited to the difference between the peak current at the shortest cycle length and that at the longest, as indicated in Figure 4A (leftmost and rightmost red bars). Figure 4A also shows that the peak current is significantly larger in the presence of forskolin at each cycle length tested (red vs. green bars), as may be anticipated from the increase in current amplitude upon forskolin observed in response to square voltage-clamp steps (Figure 1). The current during diastolic depolarization is also significantly larger in the presence of forskolin at each cycle length tested, except 1500 ms (*p* = 0.06) (Figure 4B, red vs. green bars), with a more distinct cycle length dependence (Figure 4B, red bars). At all cycle lengths, the current during diastolic depolarization was clearly above the zero current level (Figure 3, E and F; Figure 4B), indicating that the KCNQ1/KCNE1 current is continuously active during the human SAN-like AP.

### 2.3. Data on KCNQ1/KCNE1 Current Incorporated into Human SAN Cell Model

The data of Figure 2, Figure 3 and Figure 4 underscore the importance of carrying out AP clamp experiments in addition to traditional square-step voltage clamp experiments. In particular, the AP clamp experiments revealed that KCNQ1/KCNE1 current may be active during the diastolic depolarization phase, with an amount dependent on cycle length and cAMP level. However, despite the demonstration by the AP clamp measurements that KCNQ1/KCNE1 current may flow during both AP and diastolic depolarization, the net effect of the KCNQ1/KCNE1 current on AP and diastolic depolarization, and consequently, the cycle length of a free-running human SAN cell remains unknown. To study the dynamics of the KCNQ1/KCNE1 current as the *KCNQ1*/*KCNE1* encoded I_Ks_ in a human SAN cell, we used the Fabbri–Severi model of an isolated human SAN cell [41]. To this end, the original kinetics of the model I_Ks_ were replaced with those derived from our voltage clamp data on the KCNQ1/KCNE1 current, together with several other modifications to I_Ks_, also based on these data, as set out below.

To start, we adopted the second-order Hodgkin-and-Huxley-type kinetic scheme [43] as used to describe the (de)activation of I_Ks_ in the Fabbri–Severi model and in multiple other models of mammalian SAN cells, including those by Zhang et al. [44], Kurata et al. [45], Maltsev and Lakatta [28], Chandler et al. [23], Kharche et al. [46], Tao et al. [47], and Severi et al. [48]. Accordingly, I_Ks_ was described by:I_Ks_ = x_s_^2^ × g_Ks_ × (V_m_ − E_Ks_),(1)
in which x_s_ denotes the I_Ks_ gating variable, g_Ks_ denotes the I_Ks_ fully activated conductance, V_m_ denotes the membrane potential, and E_Ks_ denotes the I_Ks_ reversal potential.

To determine the V_m_-dependent activation and deactivation rate constants α and β of x_s_, respectively, we first replotted the experimental data of Figure 1C on the normalized KCNQ1/KCNE1 current by taking their square root. The resulting data obtained under control conditions (Figure 5A, green-filled circles with error bars) could be wellfitted (*r*^2^ > 0.99) by a Boltzmann curve with a half-activation voltage of −15.733 mV and a slope factor of 27.77 mV (Figure 5A, solid green line). A negative shift of this curve by 14.568 mV, thus arriving at a Boltzmann curve with a half-activation voltage of −30.301 mV and a slope factor of 27.77 mV, produced a proper fit (*r*^2^ > 0.99) to the data obtained in the presence of forskolin (Figure 5A, red-filled circles with error bars and solid red line). The Boltzmann curves of Figure 5A defined the steady-state value of x_s_ (x_s,∞_)—or, equivalently, α/(α + β) [43]—under control conditions and in the presence of forskolin.

Next, the experimental data of Figure 1E on the time constant of deactivation of the KCNQ1/KCNE1 current were inverted and halved to obtain data on the rate of deactivation, i.e., β. This rate of deactivation was highly similar under control conditions and in the presence of forskolin (Figure 5B; green- and red-filled circles with associated error bars, respectively) and could be well-fitted by a mono-exponential function (Figure 5B, solid line labeled ‘common fit’), defined by:β = 1.93 × exp(−V_m_/83.2)(2)
(*r*^2^ = 0.97, both under control conditions and in the presence of forskolin).

Having determined x_s,∞_ and β from the experimental data of Figure 1, C and E, the rate of activation α could be computed from x_s,∞_ (Figure 5A, Boltzmann curves) and β (Figure 5B, mono-exponential fit) as:α = [x_s,∞_/(1 − x_s,∞_)] × β.(3)

The thus-obtained rates of activation α under control conditions and in the presence of forskolin are shown in Figure 5C (solid green and red lines, respectively), together with the rate of deactivation β of Figure 5B, over the voltage range of a regular human SAN AP. As demonstrated in Figure 5D, our activation and deactivation rate constants α and β of the I_Ks_ gating variable x_s_ (dotted lines), as based on our experimental data on the KCNQ1/KCNE1 current, are widely different from the original ones of the Fabbri–Severi model (solid and dashed lines).

Furthermore, we set the K^+^:Na^+^ permeability ratio of the I_Ks_ channel to 1:0.0018, in accordance with our experimentally observed reversal potential of the KCNQ1/KCNE1 current, near −86.4 mV (Figure 1F, inset), instead of the 1:0.12 ratio that was used in the Fabbri–Severi model and resulted in an I_Ks_ reversal potential of −49.3 mV. Moreover, the I_Ks_ fully activated conductance g_Ks_ was decreased by 80% from its original value of 0.65 nS to 0.13 nS to maintain an I_Ks_ amplitude in the range of the original model I_Ks_. Finally, we limited the increase in g_Ks_ under β-adrenergic tone to 25%, in line with our experimental observations of Figure 1F, whereas the isoprenaline-induced increase in g_Ks_ amounted to 60% in the original Fabbri–Severi model.

### 2.4. Dynamics of KCNQ1/KCNE1 Encoded I_Ks_ in Human SAN Cell Model

We ran the Fabbri–Severi model with its modified, KCNQ1/KCNE1 current-based formulation of I_Ks_ to assess the net effect of the *KCNQ1/KCNE1* encoded I_Ks_ on the dynamics of an isolated human SAN pacemaker cell. These simulations were not only run under control conditions (‘no rate modulation’), but also under vagal tone, upon simulated administration of 20 nmol/L acetylcholine (‘20 nmol/L ACh’), and under β-adrenergic tone, upon simulated administration of a high level of isoprenaline (‘high Iso’). Figure 6 shows the results of these simulations.

Under vagal tone (‘20 nmol/L ACh’; Figure 6, A–C), the original I_Ks_ has only a small effect on AP duration and diastolic depolarization—and, consequently, on cycle length—of the model cell, as demonstrated by the almost overlapping solid blue and orange-dashed traces of Figure 6A, which were obtained in the presence and absence of I_Ks_, respectively. This is largely due to the small amplitude of I_Ks_ and its inward direction around the MDP because of its reversal potential near −50 mV (Figure 6B). With the KCNQ1/KCNE1-based I_Ks_ formulation (‘modified I_Ks_’), I_Ks_ is clearly outward over the entire diastolic phase (Figure 6B, inset, solid green trace), with an amplitude that approaches that of the inward diastolic I_f_ (Figure 6, B and C). Accordingly, diastolic depolarization is clearly slowed down when the original I_Ks_ formulation is replaced with the KCNQ1/KCNE1-based one, despite the much smaller g_Ks_ in the latter case. Also, the block of I_Ks_ has a more prominent effect on diastolic depolarization and cycle length (Figure 6A, orange-dashed vs. solid green trace).

Under control conditions (‘no rate modulation’; Figure 6, D–F), qualitatively similar effects can be observed. The amplitude of the KCNQ1/KCNE1-based I_Ks_ becomes somewhat larger than under vagal tone (Figure 6, B and E) due to the shorter diastolic phase available for deactivation. Yet, its effect on the cycle length is smaller than under vagal tone (Figure 6, A and D), which can, at least partly, be explained by the concomitant increase in I_f_ (Figure 6, C and F). This increase in I_f_ is not only due to the shorter time available for its deactivation, but also to the positive shift in its voltage dependence of activation as compared to vagal tone.

Under β-adrenergic tone (‘high Iso’; Figure 6, G–I), the original I_Ks_ is largely inward over the relatively short diastolic phase (Figure 6H), which has a clear depolarizing effect on MDP and shortens diastolic depolarization (Figure 6G). Furthermore, the peak outward I_Ks_ is now so large that AP duration is significantly shortened. As a result, the block of I_Ks_ substantially increases its cycle length rather than decreasing it (Figure 6G, orange- dashed vs. solid blue trace). The block of the considerably smaller but entirely outward modified I_Ks_ has an opposite effect on MDP and diastolic depolarization (Figure 6, G–I, orange-dashed vs. solid green trace), thus decreasing the cycle length (as under vagal tone and under control conditions), also because the AP duration is less significantly shortened due to the substantially smaller increase in the peak outward I_Ks_ as compared to the original I_Ks_ (Figure 6H, solid green vs. solid blue trace).

Figure 7 shows how I_Ks_ affects the beating rate at different levels of autonomic tone. Under a vagal tone (‘20 nmol/L ACh’), both original and modified I_Ks_ exhibit a slowing effect on the beating rate. However, the slowing effect of the original I_Ks_ is only marginal, reducing the beating rate from 49.5 bpm in the absence of I_Ks_ (‘no I_Ks_’) to 48.7 bpm in its presence (−1.6%), whereas the slowing effect of the modified I_Ks_ is more prominent, reducing the beating rate from 49.5 to 42.4 bpm (−14.4%). Under control conditions (‘no rate modulation’), these slowing effects are smaller, amounting to −0.2% (from 73.9 to 73.8 bpm) and −5.0% (from 73.9 to 70.2 bpm), respectively. Under β-adrenergic tone (‘high Iso’), the modified I_Ks_ is considerably larger than under control conditions (Figure 6, E and H), not only because of the much shorter diastolic phase available for deactivation, but also because of the increase in g_Ks_ upon β-adrenergic stimulation. Nevertheless, it exerts a similar slowing effect, reducing the beating rate from 115 to 110 bpm (−4.9%). However, as already demonstrated in Figure 6G, the original I_Ks_, which shows a dramatic increase upon β-adrenergic stimulation (Figure 6, E and H), speeds up the beating rate rather than decreasing it. The beating rate is increased by as much as 21% (from 115 to 140 bpm).

### 2.5. Bradycardic Effect of F279I Gain-of-Function Mutation in KCNQ1

Given the prominent slowing effect of the modified KCNQ1/KCNE1 current-based I_Ks_ on the beating rate, in particular under a vagal tone, we assessed whether the bradycardic effect of heterozygous gain-of-function mutations in *KCNQ1* can be explained by their effects on the intrinsic pacemaker activity of human SAN cells. To this end, we tested the effects of the F279I mutation on the Fabbri–Severi model cell. This mutation was selected because, unlike several other gain-of-function mutations in *KCNQ1*, complete and detailed patch-clamp data on its effects in a heterozygous state are available [49]. The mutation effects on I_Ks_ were implemented by a −14.6 mV shift in steady-state activation as well as faster activation and deactivation kinetics through a 30% decrease in the time constant of (de)activation. The results obtained thereby are presented in Figure 8.

Under a vagal tone (‘20 nmol/L ACh’; Figure 8, A–C), the heterozygous F279I mutation approximately doubles the amplitude of the wild-type I_Ks_, slowing down diastolic depolarization without substantially shortening AP duration and increasing the cycle length by 43% (Figure 8C, left two bars). The corresponding beating rate decreases from 42.4 to 29.7 bpm (−30%). Repeating these simulations with the original I_Ks_ formulation of the Fabbri–Severi model, the increase in cycle length is limited to 5.9% (Figure 8B, inset; Figure 8C, two right bars), reducing the beating rate from 48.7 to 46.0 bpm (−5.6%). Under control conditions (‘no rate modulation’) and under β-adrenergic tone (‘high Iso’), the effects of the F279I mutation on cycle length are much less pronounced (Figure 8, D–I). With the modified I_Ks_, the increase in cycle length amounts to 5.5% under control conditions and 1.4% under β-adrenergic tone, reducing the beating rate by 5.2% and 1.3%, respectively. With the original I_Ks_, the cycle length increases by 0.7% under control conditions and decreases by 3.9% under β-adrenergic tone, corresponding to changes in the beating rate by −0.7% and +4.1%, respectively.

## 3. Discussion

### 3.1. Overview

*KCNQ1/KCNE1* transfected HEK-293 cells displayed large time-dependent, I_Ks_-like outward currents in response to square voltage clamp steps. Forskolin significantly increased the fully-activated conductance of the KCNQ1/KCNE1 current by ≈23%, shifted the half-maximal activation voltage towards more negative potentials by ≈15 mV, and decreased the time constant of current activation by ≈30% (Figure 1). During AP clamp experiments with human SAN-like AP waveforms, KCNQ1/KCNE1 current was substantial during the AP but relatively small during diastolic depolarization. The amplitude of KCNQ1/KCNE1 current during the diastolic depolarization increased with decreasing cycle length and was significantly larger in the presence of forskolin (Figure 2, Figure 3 and Figure 4). Computer simulations with the Fabbri–Severi human SAN cell model, in which the original I_Ks_ equations were replaced with equations based on our voltage-clamp data (Figure 5), demonstrated that I_Ks_ reduced the intrinsic beating rate through its slowing effect on diastolic depolarization at all levels of autonomic tone (Figure 6 and Figure 7). I_Ks_ was smallest under vagal tone, with its long cycle length, and largest under β-adrenergic tone, with its short cycle length and its upregulation by cAMP, all in line with the data from our AP clamp experiments. Although I_Ks_ was more substantial during the AP than during diastolic depolarization, its shortening effect on AP duration was almost negligible at all levels of autonomic tone, so that, overall, I_Ks_ exerted a slowing effect on pacemaker activity at all levels of autonomic tone (Figure 6 and Figure 7). Gain-of-function mutations in *KCNQ1*, such as F279I, may exert a marked bradycardic effect, particularly during a vagal tone (Figure 8).

### 3.2. Computer Simulations with KCNQ1/KCNE1 Current Based I_Ks_

In the original Fabbri–Severi human SAN cell model, the I_Ks_ equations are based on those of the parent Severi–DiFrancesco rabbit SAN cell model [48], except for its steady-state activation curve and its reversal potential E_Ks_. The I_Ks_ steady-state activation curve was updated based on patch-clamp data on I_Ks_ in human embryonic cardiomyocytes (gestational weeks 5–9) from Danielsson et al. [50]. E_Ks_ was updated from the pure potassium equilibrium potential (E_K_) of −87.0 mV of the Severi–DiFrancesco parent model to −49.3 mV, as a consequence of the K^+^:Na^+^ permeability ratio of 1:0.12 of the I_Ks_ channel that was introduced in the Fabbri–Severi model.

Our experimentally observed reversal potential of the KCNQ1/KCNE1 current near −86.4 mV was only slightly less negative than the E_K_ of −87.9 mV in our experimental settings. Correspondingly, we set our K^+^:Na^+^ permeability ratio of the I_Ks_ channel to 1:0.0018, which made us arrive at an E_Ks_ of −85.7 mV with the intracellular and extracellular ion concentrations set to those of the Fabbri–Severi model. Our much more negative E_Ks_ than the −49.3 mV one of the Fabbri–Severi model made I_Ks_ a true outward current over the entire voltage range of the human SAN model AP. Similarly, Whittaker et al. [51] updated the E_Ks_ of the Fabbri–Severi model “from the potentially unphysiological value of −49 mV in the original model, to the more realistic value of ≈−75 mV” for consistency with the atrial cell model used in their study on *KCNQ1*-linked short QT syndrome. Of note, our modification of the original Fabbri–Severi I_Ks_ equations was not limited to the I_Ks_ reversal potential. We also modified its kinetics, based on our voltage clamp data on the kinetics of the KCNQ1/KCNE1 current, as set out in Section 2.3. As illustrated in Figure 5, the kinetics thus obtained are widely different from the original ones of the Fabbri–Severi model.

In 2013, Britton et al. [52] proposed a methodology for the construction and calibration of populations of computational models to study causes of intercellular variability in electrophysiological activity. This methodology was, for example, successfully employed by Paci et al. [53] to investigate the potential causes of the experimentally observed phenotypic variability of human-induced pluripotent stem cell-derived cardiomyocytes (hiPSC-CMs) in relation to the long QT syndrome type 3 and the variability in response of these hiPSC-CMs to antiarrhythmic drugs. Unfortunately, we could not use such populations of models in our computational study of I_Ks_ in human SAN cells because of the insufficient number of experimental recordings from human SAN cells required for calibration.

### 3.3. Simulating Gain-of-Function Mutations in KCNQ1

In our computer simulations with the Fabbri–Severi human SAN cell model, we demonstrated that I_Ks_ is an important determinant of its cycle length at all levels of autonomic tone if the original I_Ks_ equations are adapted to our experimental data (Figure 6 and Figure 7). Consequently, it is not very surprising that our simulations of the F279I gain-of-function mutation in *KCNQ1* resulted in bradycardia (Figure 8). Fabbri et al. [41] had already demonstrated that the R231C and V241F gain-of-function mutations in *KCNQ1* resulted in a decrease in the beating rate of their model cell, but they had to rely on experimental data obtained in homozygous expression systems at room temperature [54,55] for setting their model parameters, presumably thus considerably overestimating the mutational effects. More recently, Zhou et al. [56] investigated the phenotype of the *KCNQ1*-G229D gain-of-function mutation by incorporating a corresponding heterozygous I_Ks_ model into the Fabbri–Severi model. They observed that this mutation could lead to slowdown or even pacemaking failure of the model cell. Similarly, Whittaker et al. [51] had observed a pacemaker slowing when studying the *KCNQ1*-V307L gain-of-function mutation in the Fabbri–Severi model, using a Markov chain model of I_Ks_. Intriguingly, neither carriers of the G229D mutation nor carriers of the V307L mutation have presented with sinus bradycardia [56,57,58], in contrast with the aforementioned R231C, V241F, and F279I mutations. This may point to an overestimation of the fully activated I_Ks_ conductance under wild-type conditions with the simulation studies on the G229D and V307L mutations.

In earlier studies, the effects of gain-of-function mutations in *KCNQ1* on pacemaking were assessed with the use of rabbit SAN cell models. Hong et al. [59], who had found severe fetal and neonatal bradycardia associated with the heterozygous V141M mutation in *KCNQ1*, observed cessation of spontaneous activity in the Zhang et al. [44] rabbit SAN cell model. However, their model I_Ks_ consisted of 50% wild-type and 50% homozygously mutant V141M I_Ks_, thereby potentially overestimating the effect of the heterozygous mutation, now that most of the non-wild-type current will be carried by tetrameric I_Ks_ channels containing 1–3 mutant subunits rather than 4 mutant subunits, assuming that co-assembly of wild-type and mutant subunits occurs more or less randomly. Ki et al. [55] used the Kurata et al. [45] rabbit SAN cell model to investigate the effect of their *KCNQ1*-V241F gain-of-function mutation. They also observed cessation of spontaneous activity, but lack of heterozygous experimental data made them as well use a model I_Ks_ consisting of 50% wild-type and 50% homozygously mutant I_Ks_.

### 3.4. Clinical Observations

Sinus bradycardia is a common finding in short QT type 2 (SQT2) patients, including carriers of the V141M [59,60,61,62,63,64,65] and F279I [49] mutations. SQT2 patients show an abnormally short QT interval due to a heterozygous gain-of-function mutation in *KCNQ1* [40,57]. However, gain-of-function mutations in *KCNQ1* have also been associated with sinus bradycardia, while the affected patients showed normal or even prolonged QT intervals. Such mutations are the R231C [54] and V241F [55] mutations mentioned above.. These V241F and R231C mutations are associated with normal and prolonged QTc intervals, respectively. Henrion et al. [54] demonstrated that the loss-of-function nature of the R231C mutation in the ventricles may arise from differences in the KCNE β-subunit composition between atrial and ventricular I_Ks_ channels. All five KCNE subunits are expressed throughout the heart, but heterogeneously, and the KCNQ1 α-subunit interacts differently with each of the five KCNE β-subunits [66,67]. Consequently, a mutation in *KCNQ1* may have different or even opposing effects on I_Ks_ in atrial and ventricular tissue. For example, the *KCNQ1*-Q147R mutation shows a loss-of-function phenotype when co-expressed with KCNE1, but a gain-of-function phenotype when co-expressed with KCNE2 [68,69].

### 3.5. Limitations

In the present study, AP duration was not adapted when creating the set of AP clamp waveforms, as set out in Section 4.4, thus ignoring any rate dependence of AP duration and associated effects on activation or deactivation of KCNQ1/KCNE1 current. Another potential limitation is that the role of I_Ks_ may have been underestimated because our experimental data were obtained in HEK-293 cells, which lack a SAN cell like Ca^2+^ homeostasis that may increase I_Ks_ [70,71]. Also, the lack of experimental data on the amplitude of I_Ks_ in human SAN cells makes it difficult to determine the exact quantitative effect of (changes in) I_Ks_.

Although the Fabbri–Severi human SAN cell model is comprehensive in many respects, it also has its limitations. One such limitation is that autonomic modulation is restricted to the direct effects of vagal and β-adrenergic tone on specific ion currents or pumps (see Section 4.6). Thus, the model lacks a full signaling network that acts through Ca^2+^-cAMP-PKA and Ca^2+^/calmodulin-dependent protein kinase II (CaMKII)-driven mechanisms, as, for example, in the computational model of a human ventricular myocyte by Dai et al. [72] and, although limited to CaMKII, in the computational model of a human atrial myocyte by Onal et al. [73]. Another limitation is that the intracellular Na^+^ concentration is fixed to 5 mmol/L. Thus, the model cannot show an intracellular Na^+^ accumulation upon an increase in beating rate, which has been shown to be important in a computational model of a neuron [74] and furthermore appears to be important in SAN pacemaker activity, as demonstrated in a computational model of a murine SAN pacemaker cell [75].

## 4. Materials and Methods

### 4.1. Cell Preparations

Constructs of human KCNQ1 and KCNE1 were generated as described above [76]. QBI-HEK-293A cells (Qbiogene, Heidelberg, Germany) were transiently transfected with 1 µg wild-type *KCNQ1* cDNA and 1 µg *KCNE1* cDNA using lipofectamine (Gibco BRL, Life Technologies, Bleiswijk, The Netherlands). Transfected HEK-293 cells were cultured in a Minimal Essential Medium (MEM) supplemented with nonessential amino acid solution, 10% fetal bovine serum (FBS), 100 IU/mL penicillin, and 100 mg/mL streptomycin in a 5% CO_2_ incubator at 37 °C for 2 days. Transfected cells were identified under epifluorescence microscopy using a green fluorescence protein (GFP) as a reporter gene.

### 4.2. Data Acquisition

KCNQ1/KCNE1 currents were recorded at 36 ± 0.2 °C by the amphotericin-perforated patch-clamp technique using an Axopatch 200B amplifier (Molecular Devices, Sunnyvale, CA, USA). Cells were superfused with a solution containing (in mmol/L): NaCl 140, KCl 5.4, CaCl_2_ 1.8, MgCl_2_ 1, glucose 5.5, HEPES 5; pH 7.4 (NaOH). Patch pipettes (borosilicate glass; 1.5–2 MΩ) were filled with a solution containing (in mmol/L): K-gluc 125, KCl 20, NaCl 10, amphotericin-B 0.88, HEPES 10; pH 7.2 (KOH). Signals were low-pass filtered (cut-off frequency 2 kHz) and digitized at 2 kHz. Series resistance was compensated by ≥80%, and potentials were corrected for the estimated liquid junction potential. Voltage control, data acquisition, and data analysis were accomplished using custom software. Cell membrane capacitance (C_m_) was estimated by dividing the time constant of the decay of the capacitive transient in response to 5 mV hyperpolarizing voltage-clamp steps from −60 mV by the series resistance.

### 4.3. Square-Step Voltage Clamp Experiments

The activation and deactivation kinetics of the KCNQ1/KCNE1 current were determined by conventional, square-step voltage clamp protocols as diagrammed in Figure 1, A and D (insets). For both protocols, the holding potential was −80 mV and the cycle interval was 6 s. The voltage-dependence of the activation was determined by fitting a Boltzmann equation to determine the half-maximal activation voltage (V_½_) and slope factor (k) of the steady-state activation curve:I/I_max_ = A/{1.0 + exp[(V_½_ − V)/k]}.(4)

The time course of current activation was fitted by a mono-exponential equation to determine the time constant of activation (τ):I/I_max_ = A × [1 − exp(−t/τ)].(5)

The time course of current deactivation was fitted by a mono-exponential equation to determine the time constant of deactivation (τ):I/I_max_ = A × exp(−t/τ).(6)

Current densities were calculated by dividing current amplitudes by C_m_.

### 4.4. Action Potential Clamp Experiments

Characteristics of KCNQ1/KCNE1 current during human SAN-like AP waveforms were measured using the AP clamp technique [77,78]. A prerecorded AP waveform of a human SAN cell [24] was used to construct a set of five human SAN-like APs that were employed as command signals under voltage clamp conditions. The cycle length of the original AP waveform was 817 ms (Figure 9, pink trace), and this cycle length was varied by manipulating the diastolic depolarization rate to simulate faster and slower beating rates, ranging from 40 to 120 bpm (Figure 9, other traces).

### 4.5. Drugs

Forskolin (10 µmol/L; Sigma-Aldrich, Zwijndrecht, The Netherlands) was used to mimic β-adrenergic stimulation. At this concentration, forskolin exerts its maximal effect on the KCNQ1/KCNE1 current in HEK-293 cells, similar to the effect of 1 μM isoproterenol [79]. The effect of forskolin on KCNQ1/KCNE1 current in HEK-293 cells was assessed in paired experiments.

### 4.6. Computer Simulations

The CellML code of the Fabbri–Severi model, as available from the CellML Model Repository [80] at https://www.cellml.org/ (accessed on 11 March 2023), was edited and run in version 0.9.31.1409 of the Windows-based Cellular Open Resource (COR) environment [81]. All simulations were run for a period of 100 s, which appeared long enough to reach steady-state behavior. The analyzed data are from the final three seconds of the 100-s period. Vagal tone was simulated by setting the model concentration of acetylcholine (ACh) to 20 nmol/L, which mainly exerts its effects by activation of the ACh-activated potassium current I_K,ACh_, which is zero in the default model, and the inhibition of I_f_ through a negative shift in its voltage dependence [41]. To simulate the β-adrenergic tone, we shifted the voltage dependence of the I_f_ kinetics by +10 mV, increased the maximal activity of the Na^+^/K^+^ pump as well as the I_Ks_ fully-activated conductance by 60%, increased I_CaL_ through changes in its permeability (+64%) and its activation kinetics (−10.67 mV shift in its half-activation voltage and 29.244% decrease in its slope factor), shifted the voltage dependence of the I_Ks_ kinetics by −18.7 mV, and increased the SERCA pump activity by 25%. These changes are an intermediate between the changes used by Fabbri et al. [41] in these parameters to simulate the administration of 1 µmol/L of isoprenaline and their changes in these parameters used to arrive at a pacemaking rate near 180 bpm. Of note, the I_Ks_ fully activated conductance was increased by 25% instead of 60% in the modified I_Ks_ equations that we based on our experimental observations.

### 4.7. Statistics

Data are presented as mean ± SEM. Statistical analysis was carried out using Sigma-Stat, version 3.5 (Systat Software, Inc., San Jose, CA, USA). Two-way Repeated Measures ANOVA, followed by pairwise comparison using the Student–Newman–Keuls test, was used to compare the properties of the KCNQ1/KCNE1 current assessed with square-voltage clamp steps in the same seven HEK-293 cells in the absence and presence of 10 µmol/L forskolin as one factor and at various membrane potentials as the other factor (Figure 1, B, C, E and F). Similarly, we compared the KCNQ1/KCNE1 current amplitude measured with an AP clamp in the same six HEK-293 cells in the absence and presence of 10 µmol/L forskolin as one factor and at various cycle lengths as the other factor (Figure 4, A and B). A paired *t*-test was used to compare the half-maximal activation voltage (Figure 1C, inset, left two bars), the slope factor of the steady-state activation curve (Figure 1C, inset, right two bars), the reversal potential (Figure 1F, top left inset), and the fully-activated conductance (Figure 1F, bottom right inset) of the KCNQ1/KCNE1 current in the same seven cells in the absence and presence of 10 µmol/L forskolin. The level of significance was set at *p* < 0.05.

## 5. Conclusions

I_Ks_ is active during human SAN pacemaker activity at all levels of autonomic tone, but most substantially under vagal tone. Thus, it may help to extend the dynamic range of heart rate, while at the same time preventing excessive heart rates by its slowing effect on diastolic depolarization. The sinus bradycardia that is frequently observed in heterozygous gain-of-function mutations in *KCNQ1* can be explained by the slowing effect of such mutations on the intrinsic pacemaker activity of human SAN pacemaker cells, in particular under a vagal tone.

## Figures and Tables

**Figure 1 ijms-24-07264-f001:**
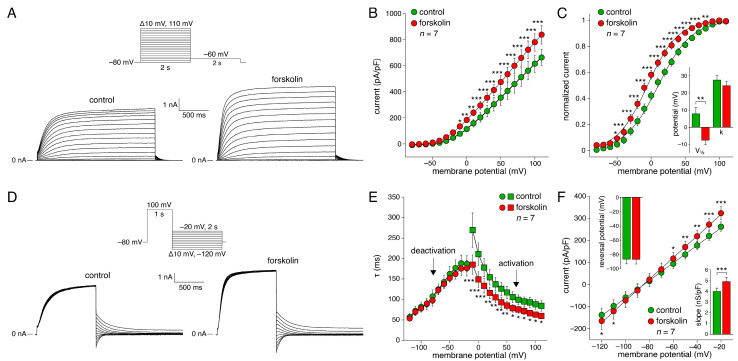
KCNQ1/KCNE1 current measured with conventional square voltage clamp steps in absence and presence of 10 µmol/L forskolin. (**A**) Typical current traces in absence (control; **left**) and presence (**right**) of forskolin. Inset: voltage clamp protocol used. (**B**) Average current density in absence and presence of forskolin. * *p* < 0.05, ** *p* < 0.01, *** *p* < 0.001, two-way Repeated Measures (RM) ANOVA. (**C**) Average voltage dependence of activation in absence and presence of forskolin. Solid lines are the Boltzmann fits to the experimental data. * *p* < 0.05, ** *p* < 0.01, *** *p* < 0.001, two-way RM ANOVA. Inset: half-maximal activation voltage (V_½_) and slope factor (k) of the individual Boltzmann curves; ** *p* < 0.01, paired *t*-test. (**D**) Typical current traces in absence and presence of forskolin. Inset: voltage clamp protocol used. (**E**) Average activation and deactivation time constants (τ) in absence and presence of forskolin. * *p* < 0.05, ** *p* < 0.01, *** *p* < 0.001, two-way RM ANOVA. (**F**) Average fully activated current in absence and presence of forskolin. Solid lines are the linear fits to the experimental data. * *p* < 0.05, ** *p* < 0.01, *** *p* < 0.001, two-way RM ANOVA. Insets: reversal potential (**top left**) and fully activated conductance (slope; **bottom right**), as determined from the individual linear fits; *** *p* < 0.001, paired *t*-test.

**Figure 2 ijms-24-07264-f002:**
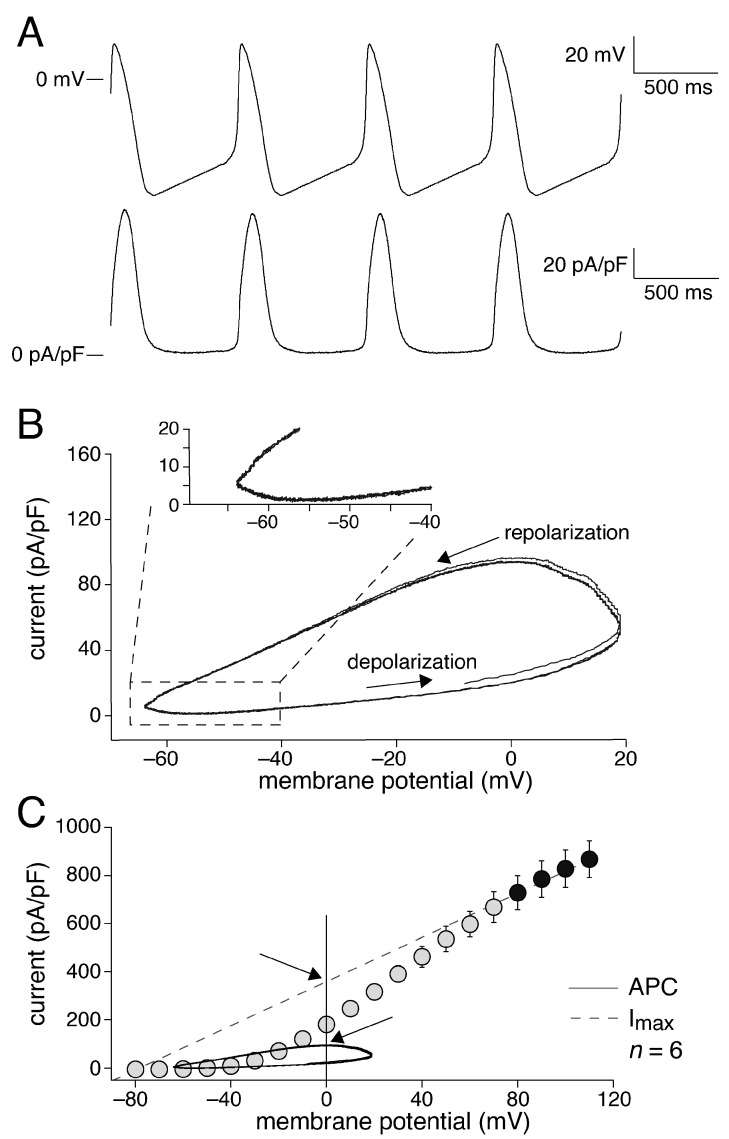
KCNQ1/KCNE1 current measured with action potential clamp. (**A**) Train of human SAN-like action potentials (APs) with a cycle length of 750 ms used as command potential in the AP clamp experiment (**top**) and average KCNQ1/KCNE1 current (*n* = 6) in response to this train of APs (**bottom**) in absence of forskolin. (**B**) KCNQ1/KCNE1 current of panel A as a phase-plane plot with arrows indicating the time course and phases of the APs. The inset shows the phase-plane plot on an enlarged current scale, focusing on the diastolic depolarization. (**C**) Static and dynamic characteristics of the KCNQ1/KCNE1 current shown as the fully activated current (I_max_; dashed line), constructed by extrapolating the currents measured at +80 mV and more positive (black-filled circles) to the zero current level near −86 mV, the steady-state current at potentials ranging from −80 to +70 mV (gray-filled circles), and the current during AP clamp (APC, solid trace) of the same six cells. The arrows relate the maximum current flowing during the human SAN-like AP waveform to the fully activated current at the associated membrane potential.

**Figure 3 ijms-24-07264-f003:**
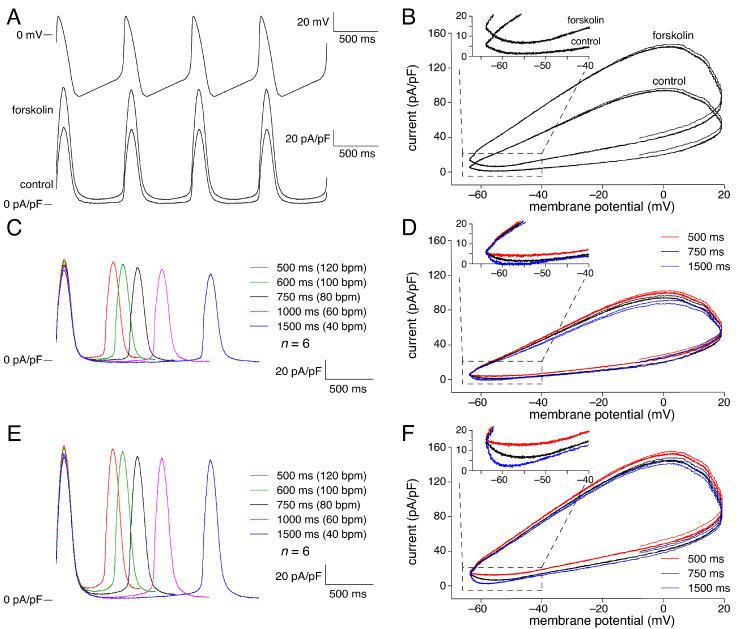
Effect of cycle length on KCNQ1/KCNE1 current in absence and presence of 10 µmol/L forskolin. (**A**) Average KCNQ1/KCNE1 current (*n* = 6) in absence and presence of 10 µmol/L forskolin (**bottom**) during a train of human SAN-like APs with a cycle length of 750 ms (**top**). (**B**) Associated phase-plane plots with inset focusing on diastolic depolarization. (**C**) Average KCNQ1/KCNE1 current in absence of forskolin at cycle lengths ranging from 500 to 1500 ms. (**D**) Phase-plane plots in absence of forskolin at cycle lengths of 500, 750, and 1500 ms. Inset: focus on diastolic depolarization. (**E**) Average KCNQ1/KCNE1 current in presence of 10 µmol/L forskolin at cycle lengths ranging from 500 to 1500 ms. (**F**) Phase-plane plots in presence of 10 µmol/L forskolin at cycle lengths of 500, 750, and 1500 ms. Inset: focus on diastolic depolarization.

**Figure 4 ijms-24-07264-f004:**
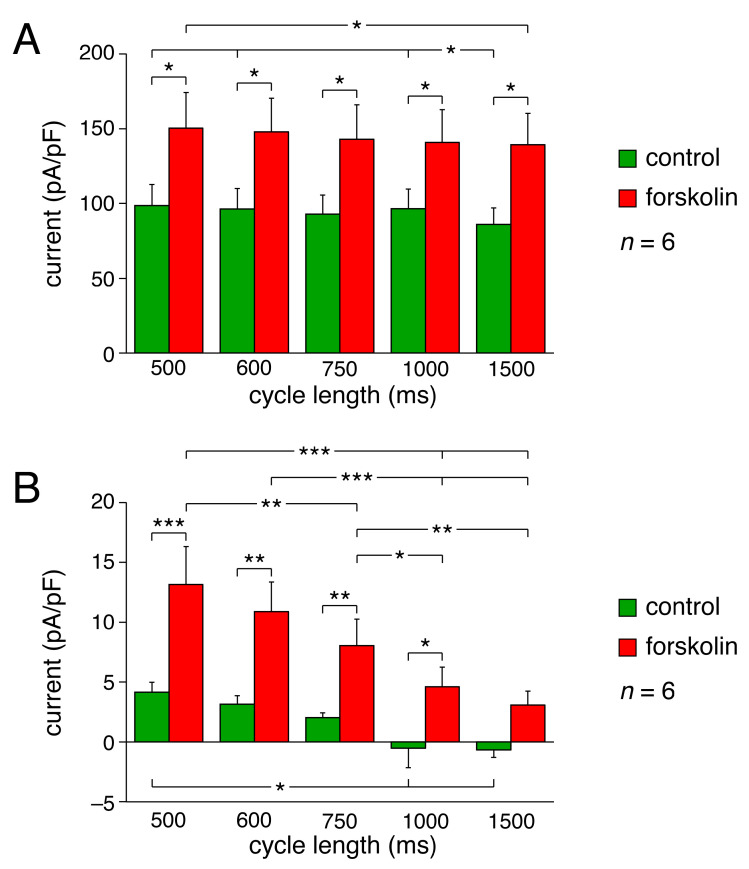
Amplitude of KCNQ1/KCNE1 current during human SAN-like APs in absence and presence of 10 µmol/L forskolin. (**A**) Peak current during AP repolarization (measured as current amplitude at 0 mV) vs. cycle length in absence and presence of forskolin. (**B**) Current during diastolic depolarization at −60 mV vs. cycle length in absence and presence of forskolin. * *p* < 0.05, ** *p* < 0.01, *** *p* < 0.001, two-way RM ANOVA.

**Figure 5 ijms-24-07264-f005:**
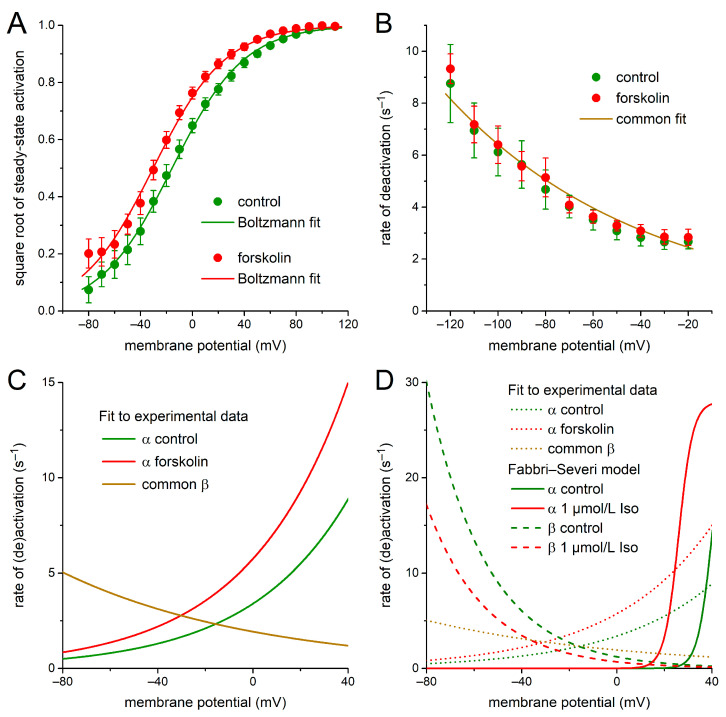
I_Ks_ kinetics derived from our voltage clamp data on KCNQ1/KCNE1 current. (**A**) Square root of the experimental data on steady-state activation of the KCNQ1/KCNE1 current in absence and presence of 10 µmol/L forskolin (filled circles) and Boltzmann fits to these data (solid lines). (**B**) Experimental data on the rate of deactivation (β) of the KCNQ1/KCNE1 current activation gate in absence and presence of 10 µmol/L forskolin (filled circles) and common mono-exponential fit to these data (solid line). (**C**) Rate of activation (α) of the KCNQ1/KCNE1 current activation gate as derived from the Boltzmann fits of panel A and the mono-exponential fit of panel B. Rate of deactivation (β) is the common fit of panel B. (**D**) Rates of activation (α) and deactivation (β) of panel C redrawn as dotted lines for comparison to the original α (solid lines) and β (dashed lines) of the I_Ks_ activation gate in the Fabbri–Severi human SAN cell model under control conditions and in presence of 1 µmol/L isoprenaline (Iso). Note difference in ordinate scale between panels C and D.

**Figure 6 ijms-24-07264-f006:**
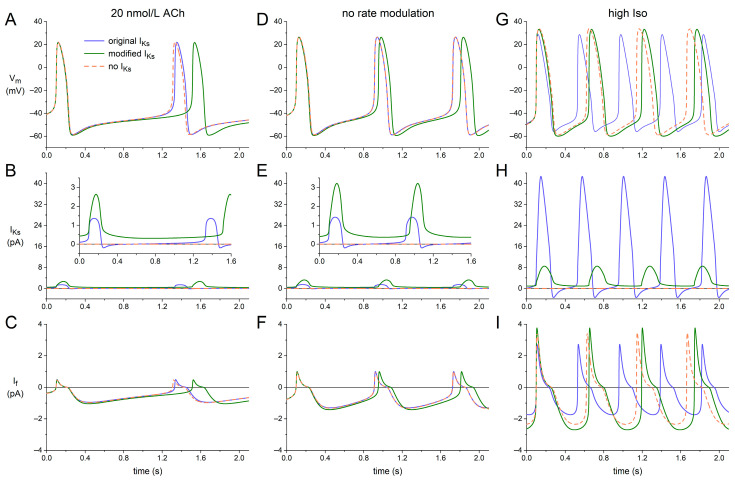
Effects of replacing the original I_Ks_ equations of the Fabbri–Severi human SAN cell model with those based on our voltage clamp data on KCNQ1/KCNE1 current. (**A**) Membrane potential (V_m_), (**B**) slow delayed rectifier potassium current (I_Ks_), and (**C**) hyperpolarization-activated ‘funny’ current (I_f_) of the Fabbri–Severi model cell with its original I_Ks_ (solid blue traces), with its modified KCNQ1/KCNE1-based I_Ks_ (solid green traces), and in the absence of I_Ks_ (orange-dashed traces) under vagal tone (20 nmol/L ACh). (**D**–**F**) V_m_, I_Ks_, and I_f_ under control conditions (no rate modulation). (**G**–**I**) V_m_, I_Ks_, and I_f_ under β-adrenergic tone (high Iso). Insets to panels B and E show I_Ks_ on an enlarged current scale.

**Figure 7 ijms-24-07264-f007:**
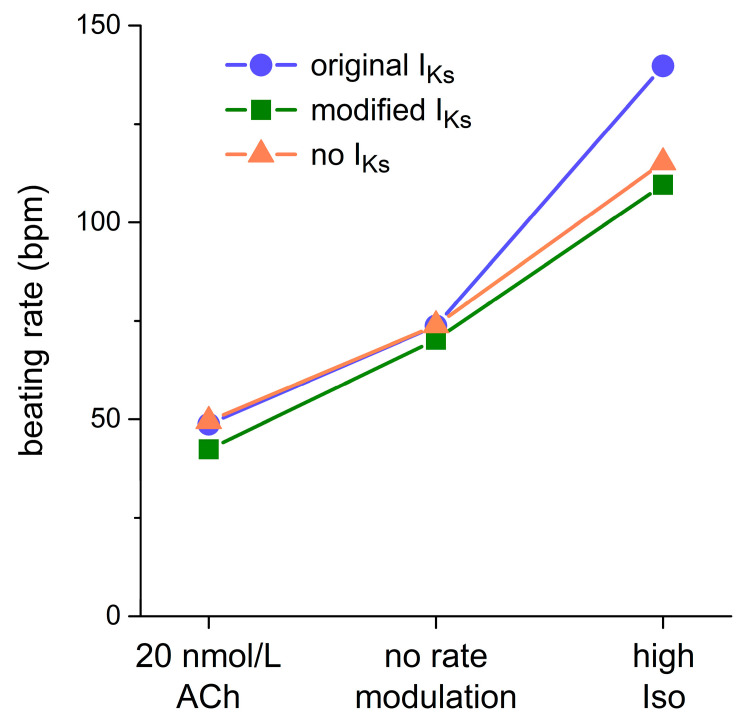
Effect of I_Ks_ on the beating rate of the Fabbri–Severi human SAN cell model. Beating rate of the Fabbri–Severi cell model with its original I_Ks_ (blue filled circles), with its I_Ks_ based on our voltage clamp data (green-filled squares), and in the absence of I_Ks_ (orange-filled triangles) under vagal tone (20 nmol/L ACh), under control conditions (no rate modulation), and under β-adrenergic tone (high Iso).

**Figure 8 ijms-24-07264-f008:**
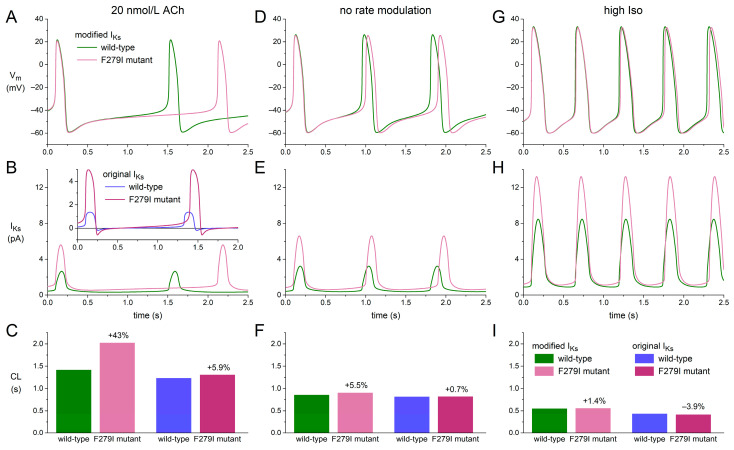
Effects of the heterozygous F279I mutation in *KCNQ1* on the Fabbri–Severi human SAN cell model with I_Ks_ based on our voltage clamp data. (**A**) V_m_, (**B**) I_Ks_, and (**C**) cycle length (CL) of the Fabbri–Severi model cell with wild-type and F279I heterozygous mutant I_Ks_ under vagal tone (20 nmol/L ACh). The inset to panel B and the two rightmost bars of panel C show I_Ks_ and CL, respectively, obtained with the original I_Ks_ equations of the Fabbri–Severi model. (**D**–**F**) V_m_, I_Ks_, and CL under control conditions (no rate modulation). (**G**–**I**) V_m_, I_Ks_, and CL under β-adrenergic tone (high Iso). The two rightmost bars of panels F and I show CL obtained with the original I_Ks_ equations.

**Figure 9 ijms-24-07264-f009:**
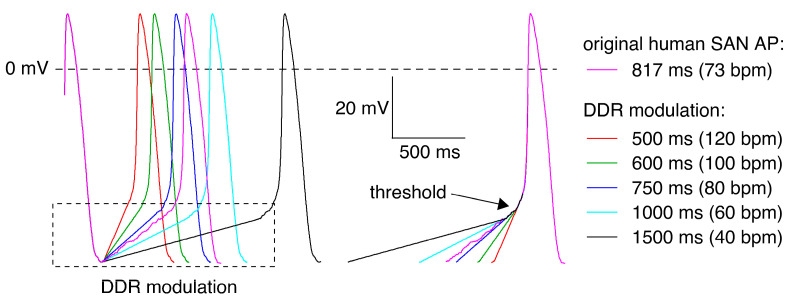
Command potentials used in AP clamp experiments. Native AP of an isolated human SAN cell (pink trace [24]) and modulation of its diastolic depolarization rate (DDR) to obtain AP waveforms with cycle lengths of 500, 600, 750, 1000, and 1500 ms (other traces), corresponding to beating rates ranging from 40 to 120 bpm.

## Data Availability

Data will be available upon request after publication to academic researchers.

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
