# Peer review of "Human Sinoatrial Node Pacemaker Activity: Role of the Slow Component of the Delayed Rectifier K+ Current, IKs"

_ijms, 2023, doi:10.3390/ijms24087264_

Round 1
Reviewer 1 Report
This paper proposed by Arie O. Verkerk et al, entitled “Human sinoatrial node pacemaker activity: role of the slow 2 component of the delayed rectifier K+ current, IKs” they assessed the role of the slowly activating component of the delayed (IKs) in human SAN pacemaker activity and its dependence on heart rate and β-adrenergic stimulation. The transfected HEK-293 cells displayed large IKs-like outward currents in response to depolarizing voltage clamp steps. Further they measured KCNQ1/KCNE1 current during human SAN-like AP waveforms in absence and presence of forskolin, and at various cycle lengths. This paper is novel and interesting aspect of cardiology. The authors have provided good evidence to support their conclusion with well-constructed and meticulously written manuscript.
However, few points must be addressed -
1) The introduction should be rewritten because it is difficult to understand for readers. I appreciate that they have provided so much information, but it must be concise.
2) In lines 35 and 36 of the introduction, the author mentions that pacemaker activity of the sinoatrial node (SAN) has been extensively studied in animals but not in "man." It is appropriate to refer to it as human. (In abstract line 2 also it needs to be changed).
3) Please elaborate statistical section on the methodology.
4) Figure 5D legends should be mentioned clearly for dotted lines.
5) A more concise presentation of the limitations section is required.
Author Response
We thank the reviewer for his/her time and efforts to review our manuscript and his/her constructive comments. We took the reviewer’s comments to heart and made changes to the manuscript accordingly. Our responses to each of the reviewer’s specific comments are given below, repeating each of the reviewer’s comments in bold, followed by our response. Changes made to the manuscript are detailed here and appear in the revised manuscript as ‘tracked changes’ through the ‘Track Changes’ function of MS Word, as requested by the editors.
This paper proposed by Arie O. Verkerk et al, entitled “Human sinoatrial node pacemaker activity: role of the slow component of the delayed rectifier K+ current, IKs” they assessed the role of the slowly activating component of the delayed (IKs) in human SAN pacemaker activity and its dependence on heart rate and β-adrenergic stimulation. The transfected HEK-293 cells displayed large IKs-like outward currents in response to depolarizing voltage clamp steps. Further they measured KCNQ1/KCNE1 current during human SAN-like AP waveforms in absence and presence of forskolin, and at various cycle lengths. This paper is novel and interesting aspect of cardiology. The authors have provided good evidence to support their conclusion with well-constructed and meticulously written manuscript.
However, few points must be addressed -
1) The introduction should be rewritten because it is difficult to understand for readers. I appreciate that they have provided so much information, but it must be concise.
We cut out large parts of the Introduction to make it more concise and focusing on the actual research presented in the manuscript. We refer to the ‘tracked changes’ of the revised manuscript for the parts that were cut out.
2) In lines 35 and 36 of the introduction, the author mentions that pacemaker activity of the sinoatrial node (SAN) has been extensively studied in animals but not in "man." It is appropriate to refer to it as human. (In abstract line 2 also it needs to be changed).
We replaced “man” with “humans” at lines 35 and 36 of the Introduction and elsewhere in the revised manuscript, including line 2 of the Abstract.
3) Please elaborate statistical section on the methodology.
We elaborated our statistical methodology in the Statistics section (Section 4.6) of the revised manuscript.
4) Figure 5D legends should be mentioned clearly for dotted lines.
In the revised manuscript, we added the dotted lines to the legend that appears in panel D of Figure 5. Furthermore, we rephrased the legend of Figure 5 to better explain the dotted lines appearing in panel D.
5) A more concise presentation of the limitations section is required.
We made our original Limitations section more concise, reducing it from three paragraphs of 20 lines in total to a single paragraph of 7 lines, which appears at the start of the revised Limitations section (Section 3.5, lines 644–650). At the same time, however, we had to expand our Limitations section with a separate paragraph (lines 651–662) in response to comments made by Reviewer #2.
Reviewer 2 Report
This manuscript used an experimental data-constrained IKs model to update a previous human SA node model. The authors explored how IKs affect the beating rate under different conditions. Overall, this manuscript is well written.
Major
1, it is a pity that the regulatory effects of PKA and Ach on ion currents are fixed. This work will be largely improved if the authors can build a β-Adrenergic signaling Network (see https://doi.org/10.1155/2016/4576313). At least this should be discussed and treated as a limitation.
2, Can the model simulations show intracellular calcium or sodium accumulation when the beating rate was increased? (see Na+ accumulation caused excitability reduction in neurons https://doi.org/10.1073/pnas.2105795118). If so, pumps such as the Na/K pump should be critical in regulating the beating rate.
3, This work falls within the scenario of using an average model to make average predictions. Individual cells are always different; therefore, the role of IKs on cellular beating rate changes may be cell-dependent (see https://doi.org/10.7554/eLife.48890 and https://doi.org/10.1073/pnas.2219049120).
For the above points, it will take a long time if the authors take those factors into account. It may be more realistic to cite relevant papers and treat them as limitations or future work.
Minor
4, Line 44-45, it is unclear what the authors meant by “expression pattern was at least appropriate to explain pace making.”
5, The authors may consider moving lines 83-85 into the Discussion. Otherwise, the statement is confusing before showing IKs is active during the diastolic depolarization phase.
6, line 11, replace “man” with “human” here and elsewhere.
Author Response
We thank the reviewer for his/her time and efforts to review our manuscript and his/her constructive comments. We took the reviewer’s comments to heart and made changes to the manuscript accordingly. Our responses to each of the reviewer’s specific comments are given below, repeating each of the reviewer’s comments in bold, followed by our response. Changes made to the manuscript are detailed here and appear in the revised manuscript as ‘tracked changes’ through the ‘Track Changes’ function of MS Word, as requested by the editors.
This manuscript used an experimental data-constrained IKs model to update a previous human SA node model. The authors explored how IKs affect the beating rate under different conditions. Overall, this manuscript is well written.
Major
1, it is a pity that the regulatory effects of PKA and Ach on ion currents are fixed. This work will be largely improved if the authors can build a β-Adrenergic signaling Network (see https://doi.org/10.1155/2016/4576313). At least this should be discussed and treated as a limitation.
2, Can the model simulations show intracellular calcium or sodium accumulation when the beating rate was increased? (see Na+ accumulation caused excitability reduction in neurons https://doi.org/10.1073/pnas.2105795118). If so, pumps such as the Na/K pump should be critical in regulating the beating rate.
3, This work falls within the scenario of using an average model to make average predictions. Individual cells are always different; therefore, the role of IKs on cellular beating rate changes may be cell-dependent (see https://doi.org/10.7554/eLife.48890 and https://doi.org/10.1073/pnas.2219049120).
For the above points, it will take a long time if the authors take those factors into account. It may be more realistic to cite relevant papers and treat them as limitations or future work.
Each of the above points is addressed in the Discussion section of the revised manuscript, as set out below, keeping in mind that Reviewer #1 requested a concise presentation of the Limitations section.
1, This point is presented as a limitation, and shortly discussed, in the second paragraph of the Limitations section of the revised manuscript (Section 3.5, lines 652–658), referring to studies with computational models of human ventricular and atrial myocytes that include such network, including the study highlighted by the reviewer.
2, The Fabbri–Severi human SAN cell model allows intracellular calcium accumulation upon an increase in beating rate, with the calcium concentrations in the sub-sarcolemma, the cytosol, and the two compartments of the sarcoplasmic reticulum as free-running variables. However, in contrast, the intracellular sodium concentration is fixed to 5 mmol/L. Thus, the model cannot show an intracellular sodium accumulation upon an increase in beating rate. The fixed intracellular sodium concentration is presented, and shortly discussed, as a limitation in the Limitations section of the revised manuscript (Section 3.5, lines 658–662), citing relevant papers, including the one brought up by the reviewer.
3, As noticed by the reviewer, we refrained from creating a population of human SAN cell models. We were unable to create such population in the absence of sufficient experimental recordings from human SAN cells required for calibration. In the revised manuscript, this point is addressed in the final paragraph of Section 3.2 (lines 563–572) of the Discussion section.
Minor
4, Line 44-45, it is unclear what the authors meant by “expression pattern was at least appropriate to explain pace making.”
Actually, we referred to Chandler et al. (2009), who concluded that the expression pattern of ion channels in the human SAN was “appropriate to explain pacemaking”. For clarification, we rephrased this sentence in the revised manuscript, now reading: “Chandler et al. [23] characterized the ‘molecular architecture’ of the human SAN based on messenger RNA (mRNA) and protein levels of 120 ion channels, and they concluded that the observed expression pattern of these ion channels is “appropriate to explain pacemaking”.”
5, The authors may consider moving lines 83-85 into the Discussion. Otherwise, the statement is confusing before showing IKs is active during the diastolic depolarization phase.
We agree and have therefore reformulated these two sentences, which appear at lines 88–93 of the Introduction section of the revised manuscript.
6, line 11, replace “man” with “human” here and elsewhere.
We replaced “man” with “humans” at line 11 (Abstract) and elsewhere in the revised manuscript, in particular in the Introduction section.
Round 2
Reviewer 2 Report
The authors solved my concerns.